

# Identification of potential biomarkers and their clinical significance in gastric cancer using bioinformatics analysis methods

Jie Liu[1], Miao Zhou[1], Yangyang Ouyang[1], Laifeng Du[2], Lingbo Xu[3] and Hongyun Li[1]

[1] Gastroenterology, Jining No.1 People's Hospital, Jining, China
[2] General Medicine, Jining Prison Hospital, Jining, China
[3] Obstetrical, Jining No.1 People's Hospital, Jining, China

Corresponding author
Hongyun Li, hongyun__li@sina.com

## ABSTRACT

**Background**. Alternative splicing (AS) is an important mechanism for regulating gene expression and proteome diversity. Tumor-alternative splicing can reveal a large class of new splicing-associated potential new antigens that may affect the immune response and can be used for immunotherapy.

**Methods**. The RNA-seq transcriptome data and clinical information of stomach adenocarcinoma (STAD) cohort were downloaded from The Cancer Genome Atlas (TCGA) database data portal, and data of splicing events were obtained from the SpliceSeq database. Predicting genes were validated by Asian cancer research group (ACRG) cohort and Oncomine database. RT-qPCR was used to analysis the expression of ECT2 in STAD.

**Results**. A total of 32,166 AS events were identified, among which 2,042 AS events were significantly associated with patients survival. Biological pathway analysis indicated that these genes play an important role in regulating gastric cancer-related processes such as GTPase activity and PI3K-Akt signaling pathway. Next, we derived a risk signature, using alternate acceptor, that is an independent prognostic marker. Moreover, high ECT2 expression was associated with poorer prognosis in STAD. Multivariate survival analysis demonstrated that high ECT2 expression was an independent risk factor for overall survival. Gene set enrichment analysis revealed that high ECT2 expression was enriched for hallmarks of malignant tumors. The ACRG cohort and Oncomine also showed that high ECT2 expression was associated with poorer prognosis in gastric cancer patients. Finally, RT-qPCR showed ECT2 expression was higher in STAD compared to the normal tissues.

**Conclusion**. This study excavated the alternative splicing events in gastric cancer, and found ECT2 might be a biomarkers for diagnosis and prognosis.

## INTRODUCTION

Gastric cancer (GC) have the second-highest mortality of cancers worldwide (*Siegel, Miller & Jemal, 2019*; *Miller et al., 2019*). With the rapid development of medical immunology and molecular biology techniques, immunotherapy as a new treatment method has received extensive attention in the field of cancer therapy. Immunotherapy is currently the most promising direction for the treatment of GC patients, however, not all GC patients are suitable for this type of approach (*Fuchs et al., 2018*; *Panda et al., 2018*; *Roh et al., 2017*). Finding the right antigen for targeted vaccines is a big challenge in people who benefit from immunotherapy (*Nishino et al., 2017*). Due to the heterogeneity of tumors, the current biomarkers for predicting prognosis have certain limitations. Therefore, this field requires new biomarkers as prognostic indicators to effectively enhance prognosis and individualized treatment.

Alternative splicing (AS) refers to the process from the precursor of mRNA to mature mRNA, in which different splicing methods enable the same gene to produce multiple different mature mRNA, and eventually produce different proteins. AS is an important mechanism for regulating gene expression and producing proteome diversity (*Nilsen & Graveley, 2010*). AS occurs frequently in tumors and is closely related to the occurrence and development of tumors (*Kim, Goren & Ast, 2008*; *Oltean & Bates, 2014*). It has been found that AS affects the family of protein genes that often mutate in tumors and changes the protein-protein interaction in tumor-related signaling pathways, indicating that AS is also an important cause of tumorigenesis (*Oltean & Bates, 2014*). Abnormal expression of splicing factors leads to changes in the variable splicing of genes (*Blencowe, 2003*), and may cause the formation of specific cancer-producing splicing isoforms, and leading to cancer (*Pradella et al., 2017*). Thus, tumor-alternative splicing can reveal a large class of new splicing-associated potential new antigens that may affect the immune response and can be used for immunotherapy.

The purpose of this study was to identify AS in GC, and to provide new splicing-associated potential new antigens on GC. Firstly, we comprehensively detected the landscape of AS events in GC. Secondly, we construct of the prognostic predictor in GC patients. Moreover, we construct survival-associated alternative splicing events. Finally, we used RT-qPCR to detect the expression of ECT2 in GC and paired adjacent normal tissue.

## MATERIAL AND METHODS

### Data acquisition

A total of 407 samples (375 GC samples and 32 normal samples) were enrolled for comprehensive integrated analysis. The data were download from The Cancer Genome Atlas (TCGA) database. In addition, we used the Data Transfer Tool (provided by GDC Apps) to download the level 3 mRNASeq gene expression data and clinical information of those patients. In addition, data of splicing events were obtained from the SpliceSeq database (*Ryan et al., 2012*). The filter condition is sample percentages with PSI values $\geq 75$ and $\Delta PSI \geq 30\%$ (*Ryan et al., 2012*). Finally, the resulting matrix files and PSI files are used for subsequent analysis.

### Gene Set Enrichment Analysis (GSEA)

GSEA abandons the previous method that the analysis software only focuses on a group of up-regulated or down-regulated genes, and focuses on a group of genes with the same or similar biological processes, through a comparative analysis of the overall changes of a group of genes, and then explain the effects of different treatments on the sample or reveal the biological significance (*Subramanian et al., 2007*). GSEA was used to enrich key Kyoto Encyclopedia of Genes and Genomes (KEGG) pathways of high and low ECT2 expression in GC.

### Oncomine database analysis

The expression level of the ECT2 gene in various types of cancers was identified in the Oncomine database (*Rhodes et al., 2007*). The threshold was determined according to the following values: *P*-value of 0.001, fold change of 1.5, and gene ranking of all.

### Quantitative reverse transcription polymerase chain reaction (qRT- PCR) assays

Total RNA from cells or tissues was isolated using TRIzol (Invitrogen, Canada) reagent, the specific operation is carried out with reference to the instructions for the operation of the kit. RNA (1 μg) was converted into cDNA using the RevertAid First Strand cDNA Synthesis Kit (Takara, China). qRT-PCR was performed using SYBR Green Mixture (Takara, China) in the ABI StepOne-Plus System (ABI7500, USA). Target gene expression was normalized against GAPDH. The primer sequences are F: 5′-CAGACTCCGAAGGAAGTTGTATG-3′,R:5′-TCCACTGAGCCGTGGGATGTCA-3′.

### Statistical analyses

We used the R packages ("UpSetR") to get a overview of AS events profiling in GC (*Conway, Lex & Gehlenborg, 2017*). We combine the survival data with the AS data to obtain survival-related AS data for subsequent analysis, then use the R package ("UpSetR and survival") to analyze the variable shear events associated with survival. A univariate Cox regression analysis was employed by "survival" package in R to identify survival-associated splicing events. Next, a multivarite Cox regression was analyzed upon meaful genes ($p < 0.05$) screened from univariate regression. Subsequently, Cytoscape 3.5 was employed to construct the potential regulatory network. R language packages (ggplot2, pheatmap, pROC, and corrgram) are used for other statistical computations and figure drawing.

## RESULT

### The landscape of AS events in gastric cancer

60,754 AS events of 22,039 genes were identified. In detail, we detected 31,730 exon skip (ES) in 6,973 genes, 8,393 alternate terminator (AT) in 3,666 genes, 10,005 alternate promoter (AP) in 4,025 genes, 4,006 alternate acceptor site (AA) in 2,799 genes, 3,450 alternate donor site (AD) in 2,401 genes, 2,944 retained intron (RI) in 1,956 genes, and 226 mutually exclusive exons in 219 genes respectively (Figs. 1A and 1B). After strict filtering by survival, 32,129 AS events of 17,913 genes were identified, including 12,894 ESs in 5,598

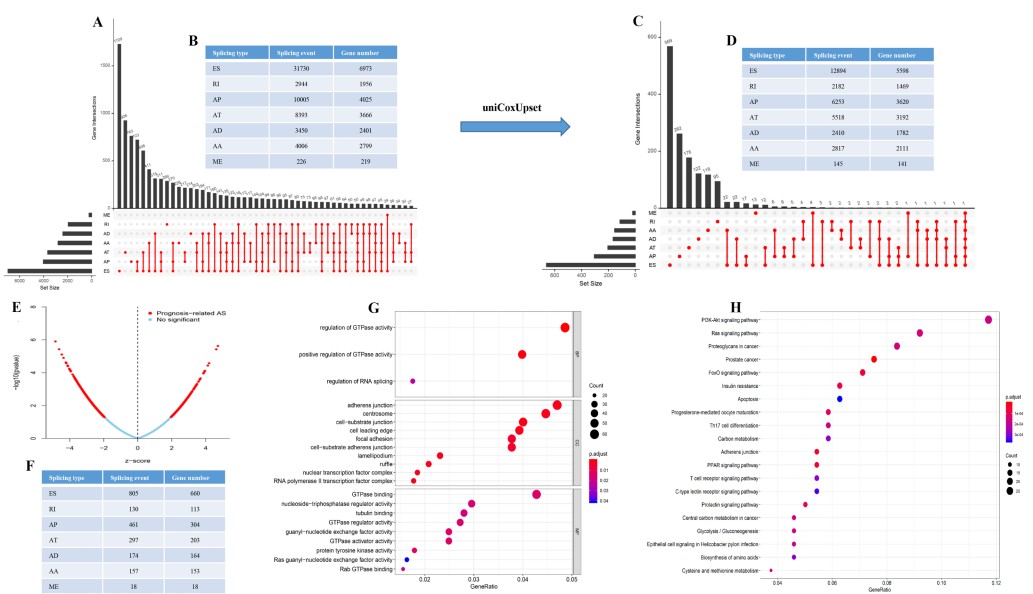

**Figure 1 The landscape of AS events in gastric cancer.** (A–B) UpSet plot of interactions between the seven types of alternative splicing events in GC. (C–D) UpSet plot of interactions between the seven types of survival associated alternative splicing events in GC. One gene may have up to five types of alternative splicing to be associated with patient survival. (E–F) Volcano plots of alternative splicing events difference for TCGA datasets. Red represents a significant difference, blue represents no significant. (G) Top 20 pathways of GO analyses of genes from OS-related alternative splicing events. Rich factor represents gene enrichment in a specific pathway. (H) Top 20 pathways of KEGG analyses of genes from OS-related alternative splicing events. Rich factor represents gene enrichment in a specific pathway. GO, Gene Ontology; KEGG, Kyoto Encyclopedia of Genes and Genomes; CC, cellular component; MF, molecular function; BP, biological process; OS, overall survival. $P < 0.05$ was statistically significant.

genes, 5,518 ATs in 3,192 genes, 6,253 APs in 3,620 genes, 2,817 AAs in 2,111 genes, 2,410 ADs in 1,782 genes, 2,182 RIs in 1,469 genes, and 145 MEs (mutually exclusive exons) in 141 genes respectively (Figs. 1C and 1D). These data suggest that a single gene may have multiple types of mRNA splicing events. ES is the primary splicing event and ME is a rare splicing event in GC patients.

To better understand the prognostic role of AS event in GC, we used a multivariate Cox regression analysis. Multivariate Cox regression showed that 2,042 AS events of 1,615 genes, including 805 ESs in 660 genes, 297 ATs in 203 genes, 461 APs in 304 genes, 157 AAs in 153 genes, 174 ADs in 164 genes, 130 RIs in 113 genes, and 18 MEs in 18 genes respectively, were significantly associated with OS ($P < 0.05$) (Figs. 1E and 1F). To better understand the biological processes of the OS-associated genes, we annotated their function using gene ontology (GO) terms and Kyoto Encyclopedia of Genes and Genomes (KEGG) pathway. The results showed that OS-associated genes are enriched in GTPase activity and PI3K-Akt signaling pathway (Figs. 1G and 1H). Obviously, most of the top 20 significant OS-associated AS events were better prognostic factors (Z < 0) (Figs. 2A–2G). For instance, AA of ECT2, LMO7, STAT3, CBX7, TRAPPC2L, TSC2, TROAP, ZNF410 and HNRNPR were adverse prognostic factors in GC patients, however others were better
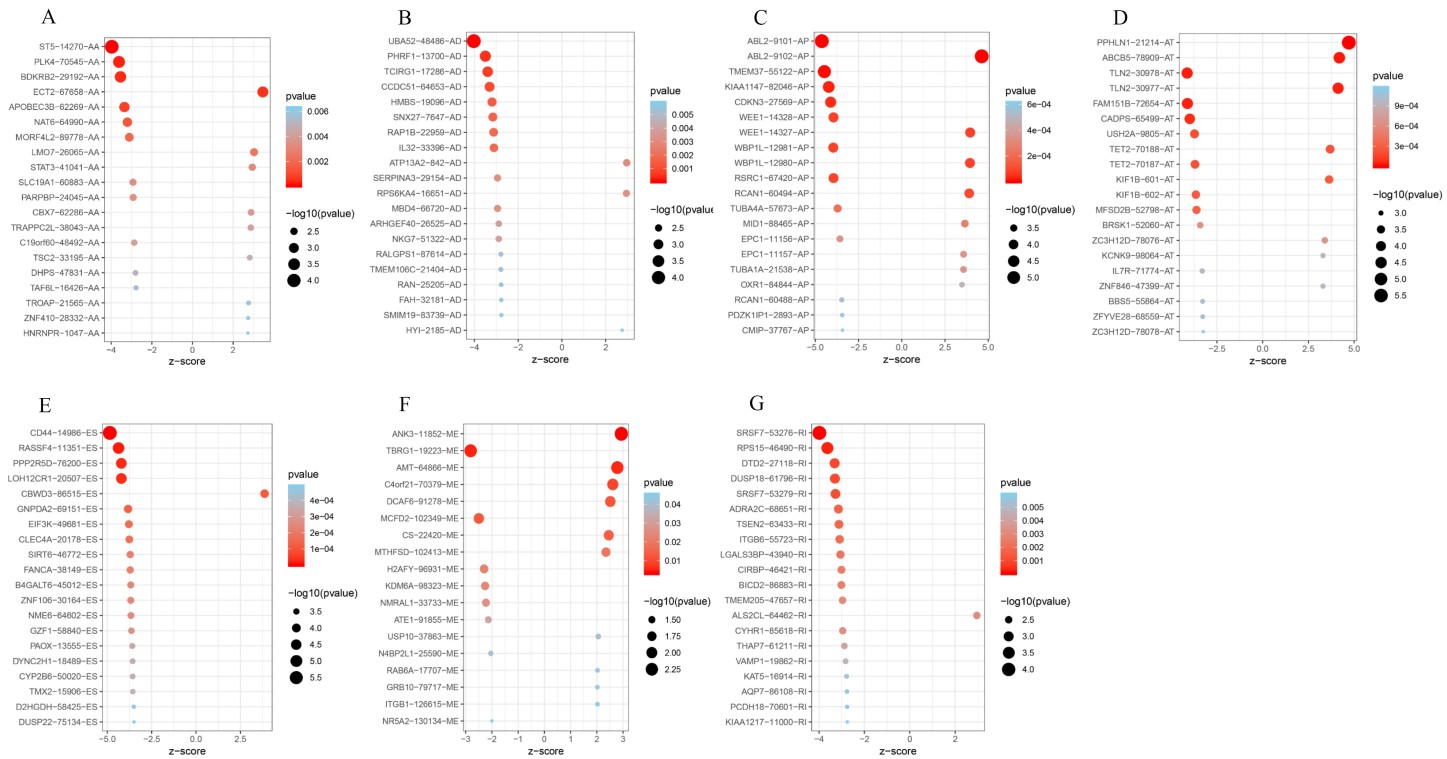

**Figure 2** **Forest plots for subgroup analyses of survival associated AS events in TCGA-STAD cohort.** (A) Forest plots of top 20 survival associated AA, AD, AP, AT, ES, ME and RI events in GC. (A) Forest plots of top 20 survival associated AA events in GC. (B) Forest plots of top 20 survival associated AD events in GC. (C) Forest plots of top 20 survival associated AP events in GC. (D) Forest plots of top 20 survival associated AT events in GC. (E) Forest plots of top 20 survival associated ES events in GC. (F) Forest plots of top 20 survival associated ME events in GC. (G) Forest plots of top 20 survival associated RI events in GC. The color scale of the circles represents p-values by the side, the larger the circle, the smaller the *P* value. Horizontal bars represent *Z* score.

prognostic factors (Fig. 2A). Table 1 shows the top 15 most significant AS events for up- and down-regulation.

## Construction of the prognostic predictor in GC patients

Next, risk score constructed using the top 20 significant OS-related AS events of the eight types, identified by multivariate Cox proportional hazards regression. As the patient's risk score increases, the number of dead patients increases, indicating that the risk score is related to survival. At the same time, as the patient's risk score increases, the PSI value of AS increases (Figs. 3A–3X). For instance, the PSI value of AP RCAN1-60494 increases as the risk value increases (Fig. 3C).

Next, the area under the curve (AUC) of ROC was generated, the result showed that risk score exhibited the AUC of 0.841 in AA, followed by AD, ES, All, ME, AP, AT and RI model with AUC of 0.827, 0.818, 0.801, 0.765, 0.759, 0.756 and 0.716, respectively (Figs. 4A–4P). The K-M curve was used to analyze the survival time of patients in the low-risk and high-risk groups. The results show that in each Cox regression model constructed from eight types of AS events, high risk score had a poor survival (Figs. 4A–4P). In addition, Both the univariate (HR: 1.71; 95% CI [1.51–1.95]) and multivariate Cox regression

**Table 1  The detailed information of the top 30 most different AS events.**

| Symbol | AS type | Z scores | HR | Lower 95% CI | Upper 95% CI | P-value |
|---|---|---|---|---|---|---|
| **Upregulated** | | | | | | |
| PPHLN1 | AT | 4.718722 | 708.4768 | 46.38922 | 10820.17 | 2.37E−06 |
| ABL2 | AP | 4.622432 | 6.340252 | 2.897372 | 13.87423 | 3.79E−06 |
| ABCB5 | AT | 4.201051 | 17.2989 | 4.575428 | 65.40413 | 2.66E−05 |
| TLN2 | AT | 4.12608 | 169.3579 | 14.79387 | 1938.783 | 3.69E−05 |
| WEE1 | AP | 3.955059 | 1530.346 | 40.41471 | 57948.17 | 7.65E−05 |
| WBP1L | AP | 3.947781 | 32.83358 | 5.801072 | 185.8353 | 7.89E−05 |
| RCAN1 | AP | 3.900007 | 9.044709 | 2.990575 | 27.35486 | 9.62E−05 |
| CBWD3 | ES | 3.865012 | 53.47146 | 7.1086 | 402.2166 | 0.000111 |
| TET2 | AT | 3.693635 | 125.1666 | 9.64922 | 1623.621 | 0.000221 |
| MID1 | AP | 3.649145 | 84.6877 | 7.805195 | 918.8761 | 0.000263 |
| KIF1B | AT | 3.64224 | 18.78038 | 3.875235 | 91.01454 | 0.00027 |
| EPC1 | AP | 3.576018 | 37.37312 | 5.136439 | 271.9296 | 0.000349 |
| TUBA1A | AP | 3.574184 | 3184.855 | 38.20425 | 265501.9 | 0.000351 |
| OXR1 | AP | 3.489226 | 8.06321 | 2.49635 | 26.04417 | 0.000484 |
| NAT6 | AA | 3.464763 | 22.01342 | 3.829565 | 126.5394 | 0.000531 |
| **Downregulated** | | | | | | |
| CD44 | ES | −4.84614 | 0.000511 | 2.38E−05 | 0.010953 | 1.26E−06 |
| ABL2 | AP | −4.62343 | 0.15761 | 0.072015 | 0.34494 | 3.77E−06 |
| TMEM37 | AP | −4.47008 | 1.17E−05 | 8.01E−08 | 0.001697 | 7.82E−06 |
| RASSF4 | ES | −4.36672 | 3.02E−06 | 1.00E−08 | 0.000906 | 1.26E−05 |
| KIAA1147 | AP | −4.22552 | 4.51E−11 | 7.16E−16 | 2.84E−06 | 2.38E−05 |
| PPP2R5D | ES | −4.20571 | 0.000117 | 1.73E−06 | 0.007964 | 2.60E−05 |
| LOH12CR1 | ES | −4.20016 | 1.19E−05 | 5.96E−08 | 0.002359 | 2.67E−05 |
| TLN2 | AT | −4.12622 | 0.005904 | 0.000516 | 0.067586 | 3.69E−05 |
| CDKN3 | AP | −4.11316 | 3.16E−08 | 8.42E−12 | 0.000118 | 3.90E−05 |
| FAM151B | AT | −4.10608 | 1.20E−08 | 1.99E−12 | 7.25E−05 | 4.02E−05 |
| UBA52 | AD | −4.02392 | 0.020525 | 0.003092 | 0.136248 | 5.72E−05 |
| CADPS | AT | −3.99005 | 0.048147 | 0.01085 | 0.213652 | 6.61E−05 |
| SRSF7 | RI | −3.98842 | 0.082065 | 0.024019 | 0.280385 | 6.65E−05 |
| ST5 | AA | −3.98063 | 0.011741 | 0.001316 | 0.104746 | 6.87E−05 |
| WEE1 | AP | −3.95514 | 0.000653 | 1.73E−05 | 0.02474 | 7.65E−05 |

**Notes.**

HR, hazard ratio; CI, confidence interval; ES, exon skip; ME, mutually exclusive exons; RI, retained intron; AP, alternate promoter; AT, alternate terminator; AD, alternate donor site; AA, alternate acceptor site.

analyses (HR: 1.64; 95% CI [1.44–1.86]) results indicated that the risk score and age were all correlated with the OS (Figs. 5A and 5B).

Next, to determine which SF is associated with AS events associated with survival in the GC, we performed a survival analysis of SF. The results showed that 26 SF was significantly associated with overall survival. In addition, the correlation between the PSI value of significant AS events and the expression of survival-related SF was investigated using the Spearman test (Fig. 5C).

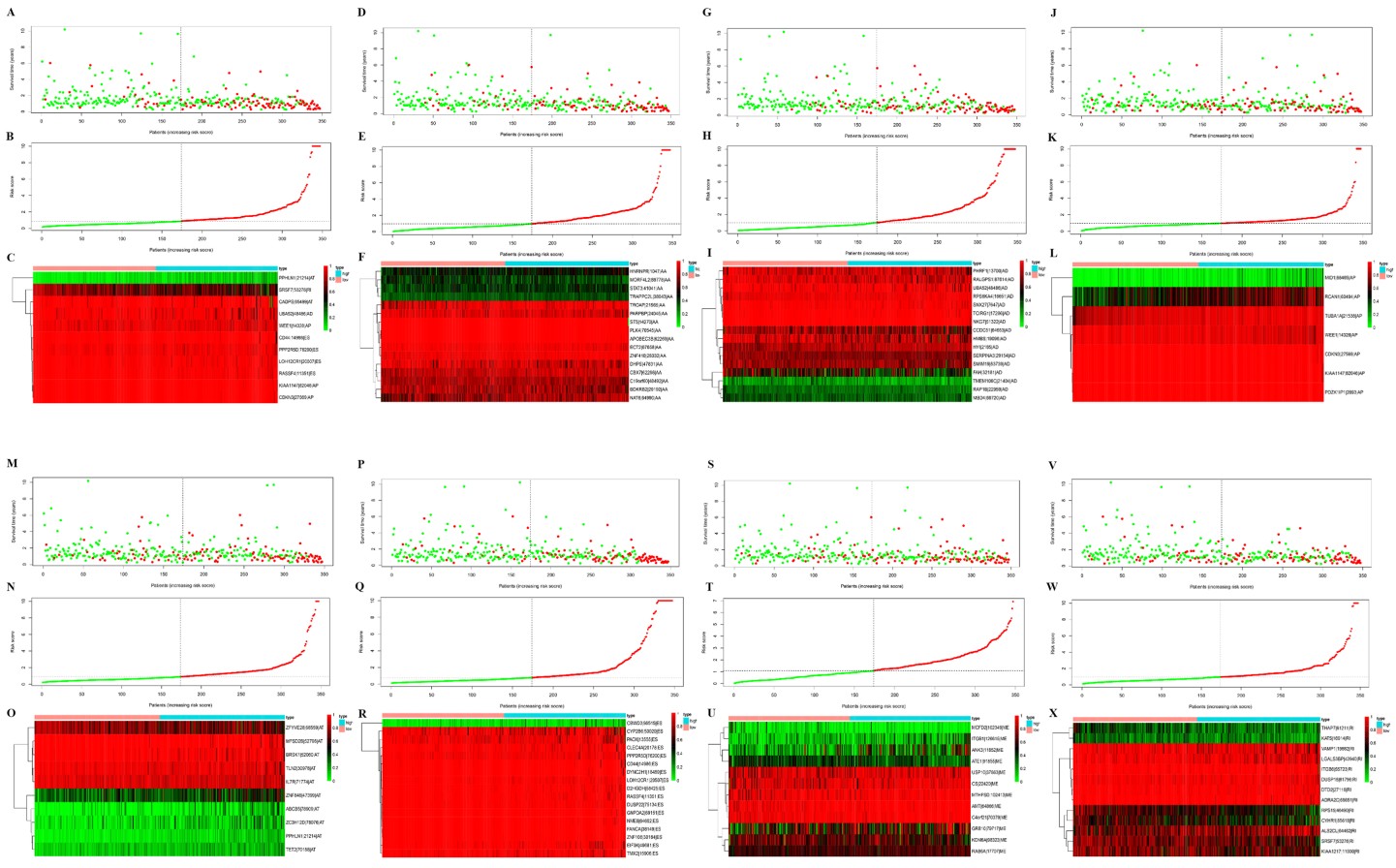

**Figure 3** **Construction and analysis of risk score based on the prognosis-associated splicing events using multiple Cox regression analysis.** GC patients were divided into low- and high-risk groups based on the median value of risk score. The top of each assembly drawing represents survival status and survival time of GC patients distributed by risk score, the middle part is the risk score curve of patients with GC, and the bottom part shows the heatmap of the PSIs for the ten splicing events. Colors from blue to red indicate increasing PSIs from low to high. (A–C) Risk scores constructed using all types of prognosis-associated splicing events. (D–F) Risk scores constructed by AA-type of prognosis-associated splicing events. (G–I) Risk scores constructed using AD-type of prognosis-associated splicing events. (J–L) Risk scores constructed using AP-type of prognosis-associated splicing events. (M–O) Risk scores constructed using AT-type of prognosis-associated splicing events. (P–R) Risk scores constructed using ES-type of prognosis-associated splicing events. (S–U) Risk scores constructed using ME-type of prognosis-associated splicing events. (V–X) risk scores constructed using RI-type of prognostic-associated splicing events.

## Construct survival-associated alternative splicing events

The counterpart genes of risk score (AA) were ST5, PLK4, BDKRB2, NAT6, APOBEC3B, ECT2, MORF4L2, STAT3, PARPBP, CBX7, TRAPPC2L, C19orf60, DHPS, TROAP, ZNF410, and HNRNPR (Table 2). Compared with normal tissues, GC patients generally contain a lower proportion of ST2 and CBX7 ($P < 0.05$), while the mRNA expression of PLK4, NAT6, APOBEC3B, ECT2, MORF4L2, STAT3, PARPBP, TRAPPC2L, TROAP, and HNRNPR were opposite ($P < 0.05$). Compared with normal tissues, GC patients have no considerable difference of BDKRB2, ZNF410, and DHPS (Figs. 6A and 6B). K-M analysis was used to evaluate the association between mRNA expression of 15 genes and

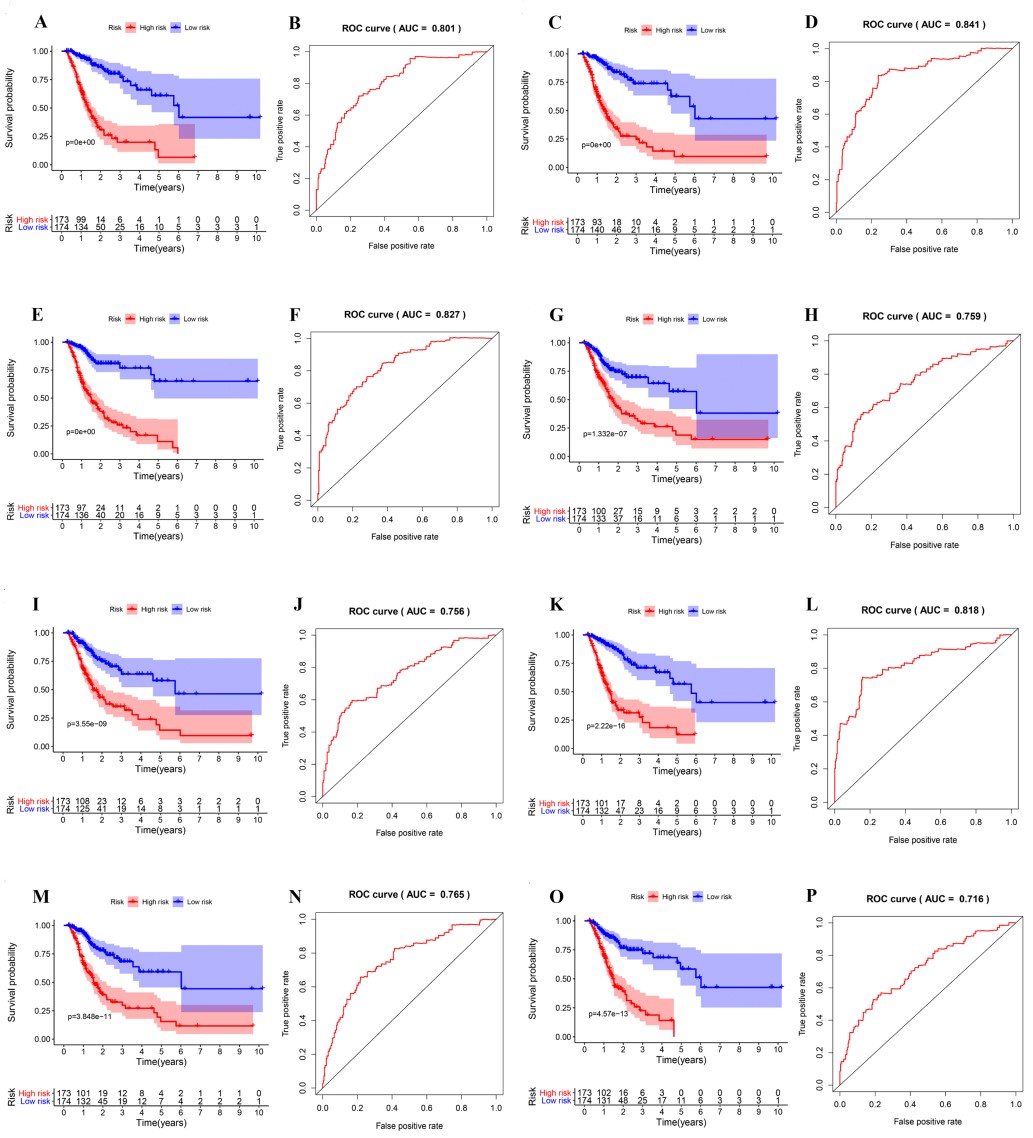

**Figure 4** **ROC and K-M curves of eight risk scores constructed using survival-associated alternative spicing events in GC.** (A–P) K-M curves of all-type, AA-type, AD-type, AP-type, AT-type, ES-type, ME-type, RI-type risk scores in GC patients, divided into low- and high-risk groups based on the median value of the risk score, and ROC curves of all-type, AA-type, AD-type, AP-type, AT-type, ES-type, ME-type, RI-type risk scores for predicting survival status of patients with GC.

OS, and the result indicate high expression of ECT2 predicts poor prognosis in GC patients (Figs. 6C–6G).

## High expression of ECT2 predicts poor prognosis in GC patients

Next, both the univariate (HR: 1.32; CI [1.11–1.56]) and multivariate Cox regression analysis (HR: 1.26; CI [1.06–1.51]) results indicated that high ECT2 expression correlated significantly with a poor overall survival (Fig. 7A). To identify signaling pathways that are

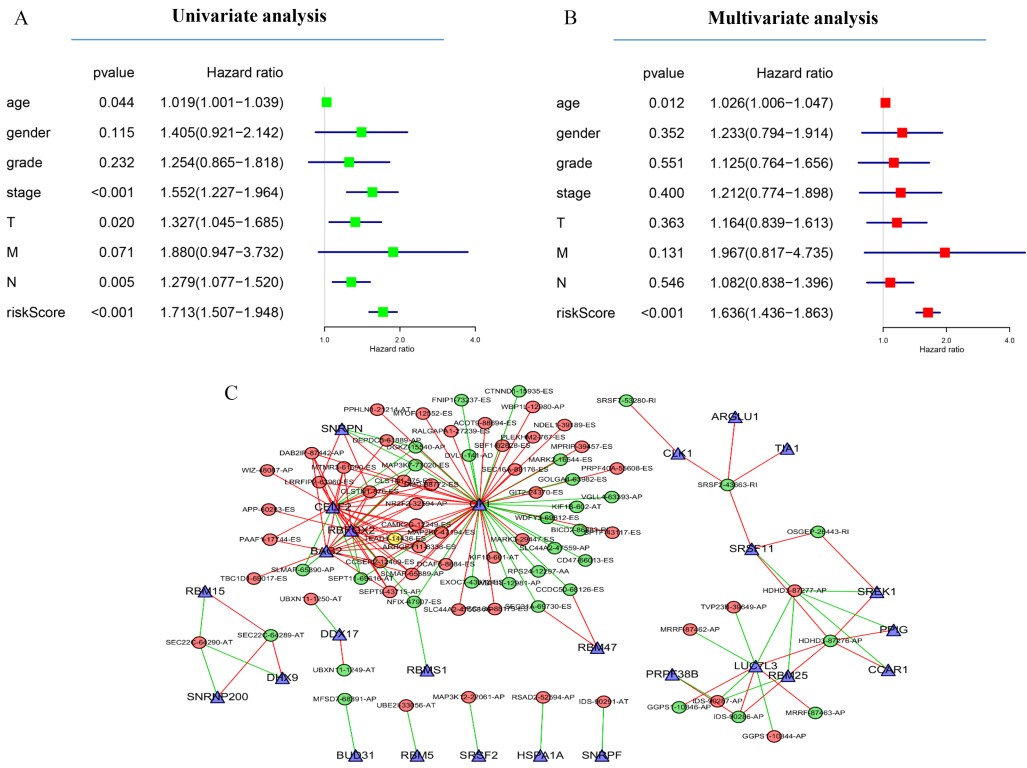

**Figure 5  Network of survival-associated AS splicing factors.** (A) Univariate Cox regression analysis of the association between clinicopathological factors (including the risk score) and OS of patients in the TCGA datasets. (B) Multivariate Cox regression analysis of the association between clinicopathological factors (including the risk score) and overall survival of patients in the TCGA datasets. (C) Correlation network between expression of survival AS factors and PSI values of AS genes generated using Cytoscape. Gray dots were survival associated splicing factors. Green/Red dots were favorable/adverse AS events. Red/green lines represent positive/negative correlations between substances.

differentially activated in GC, we conducted GSEA between low and high ECT2 expression data sets. We selected the most significantly enriched signaling pathways based on their normalized enrichment score. The GSEA shows that cancer pathway, prostate cancer pathway, and wnt signaling pathway are differentially enriched in ECT2 high expression phenotype (Figs. 7B–7F), and parkinson's disease, ribosome, oxidative, and Huntington disease are differentially enriched in ECT2 low expression phenotype (Fig. 8B). Next, we further validated ECT2 using data from Oncomine, TIMER, and GEO database. The results showed that mRNA levels of ECT2 were significantly upregulated in GC patients compared with normal samples, and high expression of ECT2 predicts poor prognosis in GC patients (Figs. 8A–8C). To further validate ECT2 in GC, RT-qPCR was used to detect the ECT2 mRNA expression in GC, and paired adjacent normal tissue (PANT). Compared with the PANT group, the ECT2 mRNA level was significantly higher in the GC group (Fig. 8D).

**Table 2  GC-specific genes involved in the ideal prognostic model.**

| Symbol | AS type | Z scores | HR | Lower 95% CI | Upper 95% CI | P-value |
|---|---|---|---|---|---|---|
| ST5 | AA | −3.98063 | 0.011741 | 0.001316 | 0.104746 | 6.87E−05 |
| PLK4 | AA | −3.62387 | 2.39E−30 | 2.28E−46 | 2.51E−14 | 0.00029 |
| BDKRB2 | AA | −3.54094 | 0.043664 | 0.007716 | 0.247079 | 0.000399 |
| ECT2 | AA | 3.464763 | 22.01342 | 3.829565 | 126.5394 | 0.000531 |
| APOBEC3B | AA | −3.35143 | 6.97E−25 | 5.20E−39 | 9.34E−11 | 0.000804 |
| NAT6 | AA | −3.21002 | 3.08E−05 | 5.41E−08 | 0.017498 | 0.001327 |
| MORF4L2 | AA | −3.11388 | 0.031742 | 0.003618 | 0.27845 | 0.001846 |
| LMO7 | AA | 3.030706 | 4.169178 | 1.656008 | 10.49635 | 0.00244 |
| STAT3 | AA | 2.944662 | 58.29749 | 3.894351 | 872.6994 | 0.003233 |
| SLC19A1 | AA | −2.9306 | 0.005836 | 0.000187 | 0.182022 | 0.003383 |
| PARPBP | AA | −2.92667 | 0.033916 | 0.003518 | 0.327025 | 0.003426 |
| CBX7 | AA | 2.889714 | 84.91815 | 4.174894 | 1727.252 | 0.003856 |
| TRAPPC2L | AA | 2.868993 | 802.8213 | 8.323711 | 77432.05 | 0.004118 |
| C19orf60 | AA | −2.86832 | 0.01071 | 0.000482 | 0.237716 | 0.004127 |
| TSC2 | AA | 2.817036 | 12.70887 | 2.167305 | 74.52356 | 0.004847 |
| DHPS | AA | −2.81067 | 0.070895 | 0.011197 | 0.448865 | 0.004944 |
| TAF6L | AA | −2.78302 | 0.000388 | 1.53E−06 | 0.097973 | 0.005386 |
| TROAP | AA | 2.765524 | 36.35456 | 2.848195 | 464.0322 | 0.005683 |
| ZNF410 | AA | 2.741289 | 152873.2 | 30.03605 | 7.78E+08 | 0.00612 |
| HNRNPR | AA | 2.733637 | 19.5677 | 2.320231 | 165.0245 | 0.006264 |

**Notes.**

HR, hazard ratio; CI, confidence interval; AA, alternate acceptor site.

## DISCUSSION

Invasion and metastasis are the important characteristics of GC, and leads to a poor prognosis. Surgery, radiotherapy, and chemotherapy are the predominant treatments for GC. Immunotherapy represented by anti-PD-1/PD-L1 monoclonal antibody drugs and CAR-T cell therapy has attracted much attention, and encouraging results have continued. Both of them are essentially the ability of human autoimmune system to recruit and activate human core immune guardian-T cells to identify and clear cancer cells through antigen-antibody response (*Le et al., 2017*). However, not every patient responds to this treatment, especially in GC (*Grosser et al., 2019*). Therefore, there is an urgent need to clarify and identify new biomaker for therapeutic target.

Previous studies suggest that AS may be associated with 50% of the human genetic diseases (*Pan et al., 2008*), including hypercholesterolemia (*Zhu et al., 2007*), frontotemporal dementia (*Ayala et al., 2005*), and tumors (*Kim, Goren & Ast, 2008*). The overall function of variable splicing is to increase the diversity of mRNA expressed from the genome, altering the protein encoded by the mRNA, and the effect of variable splicing on protein structure and function changes the phenotype (*Lara-Pezzi et al., 2017*). Studies on the different phenotypes of the same species through variable shear have positive implications for biological evolution (*Bush et al., 2017*; *Lin, Taggart & Fairbrother, 2016*). AS provides a means for cells to diversify proteomes, and there is growing evidence that

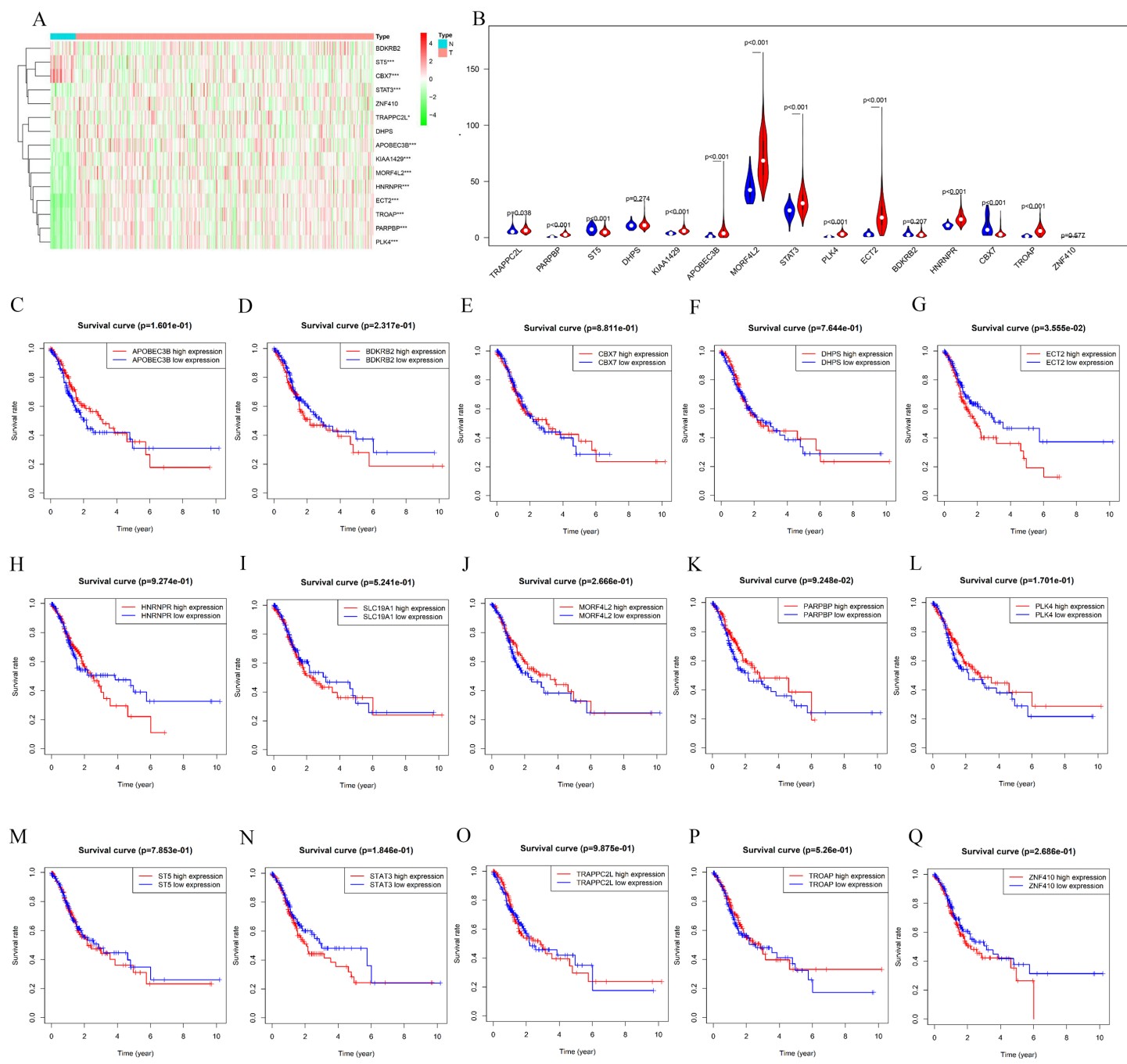

**Figure 6** **mRNA expression and K-M curves of genes from the fifteen splicing events used in constructing "risk score (AA)" in GC.** (A) The heatmap shows the expression levels of the fifteen genes in in 375 tumor patients and 32 normal tissues in TCGA dataset. (B) Vioplot visualizing the differentially fifteen genes in gastric cancer. (C–Q) K-M curves of fifteen genes in GC patients, divided into low-and high-expression groups according to the median value of mRNA expression of fifteen genes. * $P < 0.05$, ** $P < 0.01$ and *** $P < 0.001$.

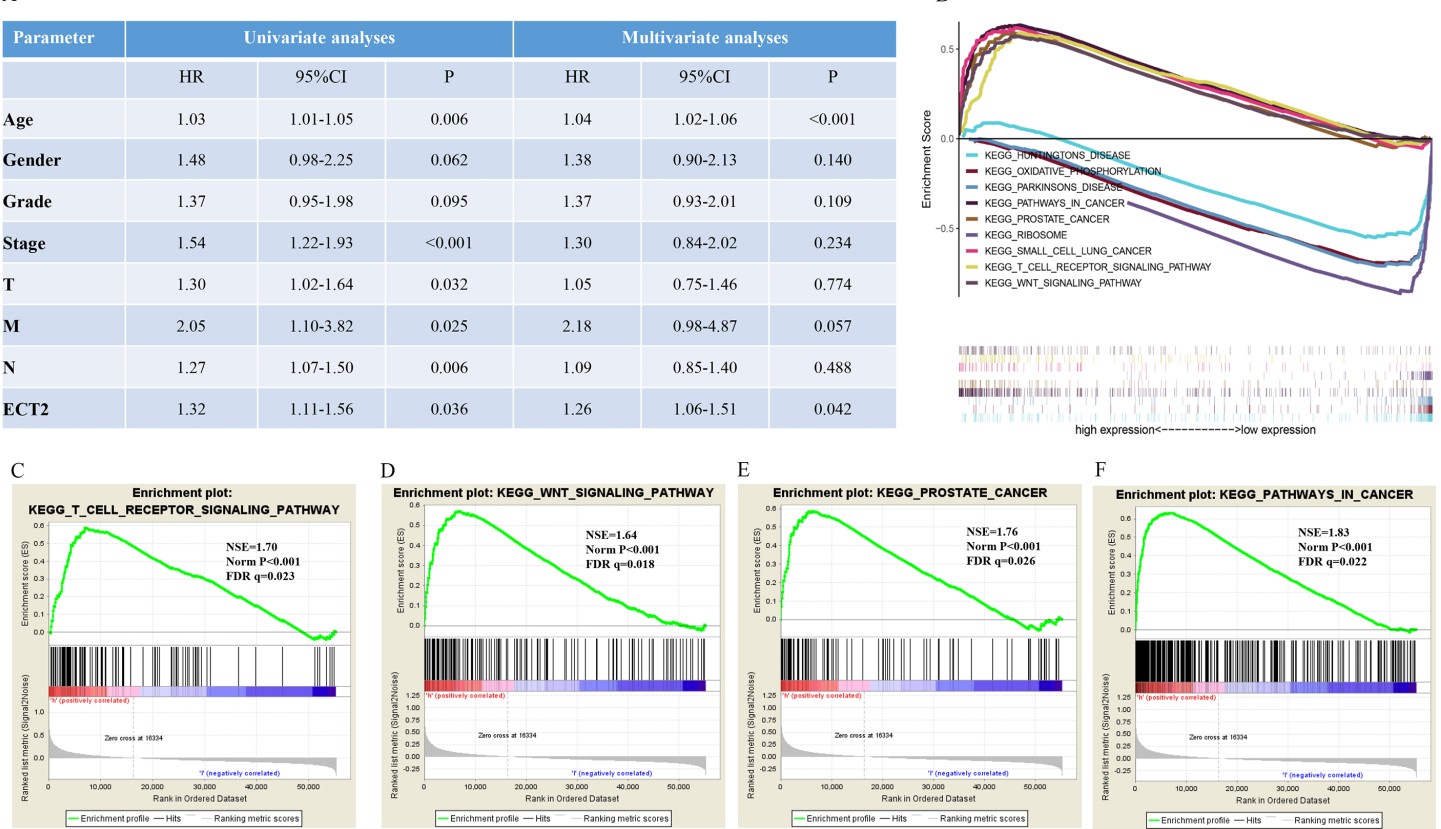

**Figure 7** **GSEA identifies a ECT2-related signaling pathway levels.** (A) Univariate and multivariate Cox regression analysis of the association between ECT2 and OS of patients in the TCGA datasets. (B) GSEA revealed that ECT2 in TCGA datasets were enriched for hallmarks of malignant tumors. GSEA results showing T cell receptor signaling pathway (C), wnt pathway (D), prostate pathway (E) and pathway in cancer (F) are differentially enriched in ECT2-related GC. ES, enrichment score; NES, normalized ES; Norm p-val: normalized *p*-value.

AS plays a key role in the development or progression of human disease, including GC (*Li & Yuan, 2017*). The expression of tumor-specific splicing variants affects many cellular activities closely related to cancer, such as cell proliferation, motility, and drug response (*Skotheim & Nees, 2007*).

This article provides analysis of the alternative splicing of genomic maps from 375 GC patients by reanalysing mRNA data. A total of 32,166 AS events were identified, among which 2,042 AS events were significantly associated with patients survival. Biological pathway analysis indicated that these SASEs play an important role in regulating gastric cancer-related processes. Next, we derived a risk signature, using alternate acceptor, that is an independent prognostic marker. Moreover, high ECT2 expression was associated with poorer prognosis in STAD. Multivariate survival analysis demonstrated that high ECT2 expression was an independent risk factor for overall survival, and as validated in GEO database. GSEA revealed that high ECT2 expression was enriched for hallmarks of malignant tumors. Finally, RT-qPCR showed ECT2 expression was higher in STAD compared to the normal tissues.

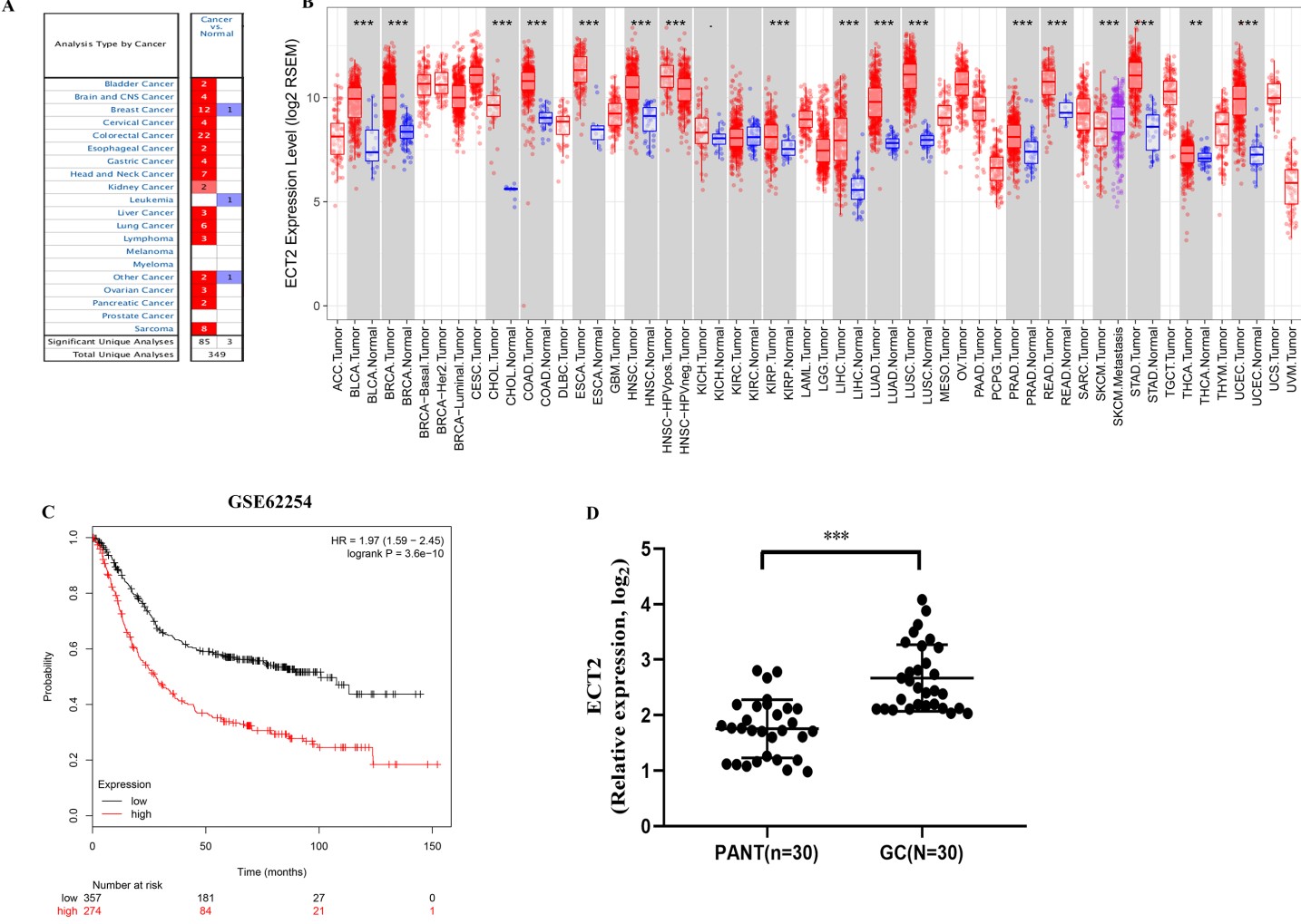

**Figure 8  ECT2 expression levels in different types of human cancers.** (A) Increased or decreased ECT2 in data sets of different cancers compared with normal tissues in the Oncomine database. (B) Human ECT2 expression levels in different tumor types from TCGA database were determined by TIMER. (C) Survival curves of OS in ACRG cohort (GSE62254). (D) ECT2 mRNA level was shown for the GC and paired adjacent normal tissue (PANT). * $P < 0.05$, ** $P < 0.01$ and *** $P < 0.001$.

The ECT2 gene, located on human chromosome 3q26, is a highly conservative gene (*Solski et al., 2004*). It can transform fibroblasts into cancer cells and interact with members of the Rho GTP family to cause malignant transformation, induce cell division and regulate the polarity of epithelial cells (*Kim et al., 2014*). ECT2 has been thought to be associated with a variety of cancers. The expression of ECT2 is closely related to cell cycle regulation and cell division. Down-regulation of ECT2 expression can block cells in G1 phase, and ECT2 expression can dynamically regulate the whole cell cycle (*Fortin et al., 2012*). Therefore, ECT2 may play a very important role in the mechanism of tumorigenesis. Whether in GC tissue or serum, the expression of ECT2 was significantly higher than that of normal controls, and the expression of ECT2 was closely related to clinicopathological parameters including tumor grade, TNM stage, and lymph node metastasis. Therefore, ECT2 plays an

important role in the occurrence and development of gastric cancer, and may be the basis for GC diagnosis and targeted therapy (*Wang, Yan & Liu, 2016*).

## CONCLUSIONS

Our study depicts a comprehensive landscape of alternative splicing events in GC and identified that SASEs can be used to predict overall survival of GC patients, and found ECT2 might be a biomarker for diagnosis and prognosis. Further investigations are needed to reveal the clinical and biological significance of the non-cancer cells and genes of alternative splicing events in GC, so as to better guide the more effective diagnosis and prognosis of gastric cancer.

### Funding
The authors received no funding for this work.

### Competing Interests
The authors declare there are no competing interests.

### Author Contributions

- Jie Liu and Miao Zhou conceived and designed the experiments, performed the experiments, analyzed the data, prepared figures and/or tables, authored or reviewed drafts of the paper, and approved the final draft.
- Yangyang Ouyang and Lingbo Xu performed the experiments, analyzed the data, prepared figures and/or tables, authored or reviewed drafts of the paper, and approved the final draft.
- Laifeng Du conceived and designed the experiments, performed the experiments, prepared figures and/or tables, authored or reviewed drafts of the paper, and approved the final draft.
- Hongyun Li conceived and designed the experiments, prepared figures and/or tables, authored or reviewed drafts of the paper, and approved the final draft.

### Data Availability
The data used to support the findings of this study are available from the cancer genome map of The Cancer Genome Atlas, TCGA (search term: TCGA-GC) and from NCBI GEO: GSE62254.

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
