# Peer review of "Identification of potential biomarkers and their clinical significance in gastric cancer using bioinformatics analysis methods"

_PeerJ, doi:10.7717/peerj.9174_

## Round 0.1 · original submission · Major Revisions

The reviewers gave very critical remarks. The paper has to be restructured. Please pay attention to the experimental validation of the predicted results by different methods, for example by RT-PCR in addition to high-throughput sequencing. However I recommend rewriting the manuscript according all the remarks. I await the revised version.

Reviewer 1 ·

Basic reporting

This is interesting study describing the identification of aberrant alternative splicing (AS) variants (32.166 in total) in 375 gastric cancer (GC) patients by using RNA-seq data and clinical information of STAD cohort from TCGA. Among them 2.042 AS events were significantly associated with patient survival. Also, pathway analysis showed that these genes play a critical role in regulating gastric cancer-related processes such as GTPase activity and PI3K-Akt signaling pathway.

1. Major concern is that the results of this manuscript are based on in silico analysis. It is important that the authors confirm some of their results by RT-PCR with RNA extracted from normal and tumor samples.
2. A number of altered AS variant were associated with CG. This is the case of CD44 exon V6; was this event also identified through analysis that the authors have performed fromTCGA?
3. In the discussion, it would be useful for the reader to improve the role of ECT2 in gastric cancer.

Minor
1) Abstract (line 24): Replace “indicated that these genes play a important role in regulating gastric” WITH “indicated that these genes play an important role in regulating gastric”.
2) Result: Page 8 (line 106): the abbreviation ME is not explained before”.

Experimental design

N/A

Validity of the findings

see above

Additional comments

see above

Reviewer 2 ·

Basic reporting

The English language should be improved to ensure that an international audience can clearly understand your text.

The discussion must be written from scratch. The version provided cannot be considered a formal discussion of results.

Experimental design

The research question is interesting but the methodological/statistical analyses require additional and mandatory controls. To note, experimental validations of the proposed results are missing. This downgrade the tone of the results.

Validity of the findings

Impact and novelty not assessed.
Statistical analyses require additional and mandatory controls.

Additional comments

Before starting with my comments, I would say that the Figures that this Reviewer can access are of very low quality and this will severely impact on my comments/concerns.

The introduction needs some more detail about the importance, and the state of art, of the therapeutical potential of targeting splicing. I would suggest that you improve this part to provide more justification for your study. Lines 63-65 provides too much details for the scope of this paper.

The Materials and methods are synthetic and require more (mandtory) details that are missing. For example, on line 83, the selection proposed by the authors is unclear. How do the authors define alternative spliced events? Line 91 did the authors correct the pvalues for multiple tests? Line 92, what kind of gene network analysis did they perform? Where are the details of GSEA analysis?

Similar to the previous section, also Results requires more details and explanations. First of all, how did the authors define AS events (line 98)? As I mentioned, Fig.1 is low quality and difficult to read. Detail on the univariate Cox model are needed. What is the stratification of samples that the authors used? Similar for the multivariate modelling (line 130)

Lines 122-127 are the obvious and not relevant.

Lines 160-170. Lauria et al (Nucleic Acids Research, 2019) have extensively shown that, when considering heterogeneous RNA-seq data (i.e. TCGA), GSEA is not the most appropriate method to perform gene set analysis. Therefore, the authors should comment on their choice of using GSEA for the analyses they performed. Details and justifications of this analysis are missing and they are required. The author never present the number of samples that they used in each analysis. This is a mandatory value to show, the reader must have the opportunity to understand whether statistical tests require or not correction for sample sizes. did the author check for this? Did they evaluate the statistical power of their results? All these analysis are missing. Justifications are required.

The Discussion is completely useless. Lines 178-245 are useless in the context of a “discussion” of the results of the paper. The only sentence that this Reviewer could link to a formal discussion of the results are lines233-234. Please write a proper discussion section where the author comments proactively on their results not on others’ results, and trash this one.

Please downgrade the tone of the conclusion section. This data must require an extensive experimental validations in the future to become real biomarkers.

The English language should be improved to ensure that an international audience can clearly understand your text. Some examples where the language could be improved include lines 80,81, 98 – the current phrasing makes comprehension difficult.

Reviewer 3 ·

Basic reporting

In this work, the authors used bioinformatics tools to analyze survival associated alternative splicing (AS) events in gastric cancer (GC) and identified the ECT2 gene as a potential biomarker.

Sufficient background information not provided as further elaborated in general comments for the authors.

Experimental design

The aim of the study is well-defined and the methods used are appropriate. However, methods are not described with sufficient details as further elaborated in general comments for the authors.

Validity of the findings

Overall conclusion are supported by the results shown. However, the impact and novelty of the findings are not assessed.

Additional comments

1) The introduction needs improvement as sufficient background covering previous bioinformatics work is not provided to understand the significance and novelty of the work. For example, lines 69-70, authors write “Integrating clinical information and large-scale of RNA-seq data to systematic analysis of AS at individual exon resolution in GC have been lacking.”. However, there has been previous bioinformatics work analyzing AS in GC (to mention a few, PMIDs: 30106437, 31857793, etc.). Consequently, the statement on line 247 “This is the most comprehensive study to assess GC predictors” should be changed accordingly.
2) The discussion needs major improvement and should be re-written. A major part of the discussion (lines 179-232) is spent describing alternative splicing or its role in cancer in general. Authors do not provide any critical analysis and interpretation of their results neither they placed their findings in the context of published literature. For example, one of the main results of the paper is the identification of ECT2 as a potential biomarker. However, previous literature (PMID: 28493890) have also identified altered AS of ECT2 in GC. Neither this result is presented in the discussion nor its importance discussed in the context of published findings.
3) Lines 243-245, “…This finding is inconsistent with previous studies that most survival-related AS events in ovarian cancer (Zhu et al. 2018). In contrast, GC showed a lower level of poor prognostic factors, but a higher level of favorable prognostic factors.…”. It is not clear what authors are trying to convey here? Moreover, what does it mean for a finding in GC data to be inconsistent with that in ovarian cancer?
4) Methods are not sufficiently detailed. For example, authors must clearly mention the dataset being analyzed (TCGA-STAD in this case), and the number of tumor and normal samples considered. Also, details of the Cox regression and clinical covariates used should be added in the methods section. Furthermore, the details of the settings used in Oncomine analysis should be added.
5) Lines 80-81, “…In RNA-seq data, splice in and splice out support splice in respectively. And the number of reads that occur with splice out.…”. It is not clear what authors are trying to convey here?
6) Lines 82-83, “…The filter condition is sample percentages with PSI values≥75, average PSI values≥0.05(…”. The authors should expand on what the filtering condition is for? Also, how can PSI >= 75 if the value range is 0-1.
7) Line 103, “…After strict filtering by survival, 32,129 AS events…”. What criteria are used for filtering?
8) There are spelling mistakes at several places in the manuscript. For example, “A amd B” on line 145. Also, abbreviations are used without first defining it. For example, STAD, ACRG, and AS on lines 18, 20, and 22 respectively.

---

## Round 0.2 · accepted · Accept

Thank you for the manuscript update. Both reviewers have no more remarks. I endorse the publication. Sorry for the delay in the manuscript processing. Some reviewers were unable to answer in time. Thanks again for submitting the work to PeerJ.

Reviewer 1 ·

Basic reporting

no comment

Experimental design

no comment

Validity of the findings

no comment

Additional comments

The authors performed most of the work that I requested and the manuscript text has been modified according to the reviewers’ suggestions
Therefore, I have no additional reservation.

Reviewer 3 ·

Basic reporting

NA

Experimental design

NA

Validity of the findings

NA

Additional comments

Authors have addressed all my concerns. I recommend the article to be considered for publication.